# GENETIC SOFT UPDATES FOR POLICY EVOLUTION IN DEEP REINFORCEMENT LEARNING

**Enrico Marchesini,**[*] **Davide Corsi, Alessandro Farinelli**
University of Verona, Department of Computer Science

## ABSTRACT

The combination of Evolutionary Algorithms (EAs) and Deep Reinforcement Learning (DRL) has been recently proposed to merge the benefits of both solutions. Existing mixed approaches, however, have been successfully applied only to actor-critic methods and present significant overhead. We address these issues by introducing a novel mixed framework that exploits a periodical genetic evaluation to soft update the weights of a DRL agent. The resulting approach is applicable with any DRL method and, in a worst-case scenario, it does not exhibit detrimental behaviours. Experiments in robotic applications and continuous control benchmarks demonstrate the versatility of our approach that significantly outperforms prior DRL, EAs, and mixed approaches. Finally, we employ formal verification to confirm the policy improvement, mitigating the inefficient exploration and hyper-parameter sensitivity of DRL.

## 1 INTRODUCTION

The key to a wider and successful application of DRL techniques in real scenarios is the ability to adapt to the surrounding environment by generalizing from training experiences. These solutions have to cope with the uncertainties of the operational environment, requiring a huge number of trials to achieve good performance. Hence, devising robust learning approaches while improving sample efficiency is one of the challenges for wider utilization of DRL. Despite the promising results (Tai et al., 2017; Zhang et al., 2017; Marchesini et al., 2019), DRL also suffer from convergence to local optima, which is mainly caused by the lack of diverse exploration when operating in high-dimensional spaces. Several studies address the exploration problem (e.g., curiosity-driven exploration (Pathak et al., 2017), count-based exploration (Ostrovski et al., 2017)), but they typically rely on sensitive task-specific hyper-parameters. The sensitivity to such hyper-parameters is another significant issue in DRL as it typically results in brittle convergence properties and poor performance in practical tasks (Haarnoja et al., 2018).

Evolutionary Algorithms (Fogel, 2006) have been recently employed as a promising gradient-free optimization alternative to DRL. The redundancy of these population-based approaches has the advantages of enabling diverse exploration and improve robustness, leading to a more stable convergence. In particular, Genetic Algorithms (GA) (Montana & Davis, 1989) show competitive results compared to gradient-based DRL (Such et al., 2017) and are characterized by low computational cost. These gradient-free approaches, however, struggle to solve high-dimensional problems having poor generalization skills and are significantly less sample efficient than gradient-based methods.

An emergent research direction proposes the combination of gradient-free and gradient-based methods following the physical world, where evolution and learning cooperate to assimilate the best of both solutions (Simpson, 1953). The first mixed approach, Evolutionary Reinforcement Learning (ERL) (Khadka & Tumer, 2018), relies on actor-critic architecture to inject information in an evolutionary population while both the gradient-free and gradient-based training phases proceed in parallel. Similarly, Proximal Distilled ERL (PDERL) (Bodnar, 2020) extends ERL with different evolutionary methods. CEM-RL (Pourchot, 2019) brings this research direction into the family of distributed approaches, combining a portfolio of TD3 (Fujimoto et al., 2018) learners with the Cross-Entropy Method (Yan Duan, 2016).

---

[*]Contact author: `enrico.marchesini@univr.it`

These mixed approaches however also present several limitations, which we address through our work: (i) the parallel training phases of the DRL and EA components (Khadka & Tumer, 2018; Bodnar, 2020), or the multitude of learners (Pourchot, 2019) result in significant overhead (detailed in Section 4). (ii) The actor-critic formalization of previous mixed approaches, allows them to be easily evaluated in continuous locomotion benchmarks (Brockman et al., 2016; Todorov et al., 2012). However, this also hinders their combination with value-based DRL (Marchesini & Farinelli, 2020a). This

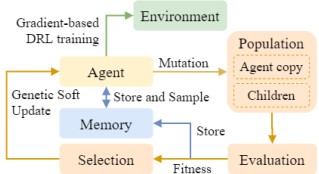

Figure 1: Supe-RL overview.

is important as recent work (Matheron et al., 2019) shows the limitation of actor-critic in deterministic tasks, that in contrast can be effectively addressed with value-based DRL. In particular, Section 4 shows that a value-based implementation of Khadka & Tumer (2018) does not converge in our discrete robotic task. (iii) The combination strategy does not ensure better performance compared to the DRL agent as it does not prevent detrimental behaviours (e.g., drop in performance). This is shown in the poor performance of a value-based implementation of ERL and PDERL (Section 4).

We propose a novel mixed framework, called Soft Updates for Policy Evolution (Supe-RL), that enables us to combine the characteristics of GAs with any DRL algorithm, addressing the limitations of previous approaches. Supe-RL (Figure 1) benefits from the high sampling efficiency of gradient-based DRL while incorporating gradient-free GA to generate diverse experiences and find better policies. Summarizing, Supe-RL based algorithms perform a periodical genetic evaluation applying GAs to the agent network. A selection operator uses a fitness metric to evaluate the population, choosing the best performing genome (i.e., the weights of the network) that is used to update the weights of the DRL agent. In contrast to previous work, our genetic evaluation is only performed periodically, drastically reducing the overhead. Furthermore, our soft update (Section 3) allows a direct integration of GAs to any DRL algorithm as it is similar to perform a gradient step towards a better policy, avoiding detrimental behaviours. As detailed in Section 3.1, this allows using value-based DRL, exploiting the variety of optimizations developed for the well-known DQN (van Hasselt et al., 2016; Schaul et al., 2016; Wang et al., 2016; Fortunato et al., 2017; Bellemare et al., 2017). Crucially, our genetic component influences the DRL agent policy only if one of its mutated version performs better in a subset of evaluation episodes. Hence, as detailed in Section 3, with a sufficient number of episodes we obtain a good estimation of the overall performance of the population.

Our evaluation focuses on mapless navigation, a well-known problem in robotics and recent DRL (Zhang et al., 2017; Wahid et al., 2019; Marchesini & Farinelli, 2020b). In particular, we consider two tasks developed with Unity (Juliani et al., 2018): (i) a discrete action space indoor scenario with obstacles for a mobile robot and (ii) a continuous task for aquatic drones, with dynamic waves and physically realistic water. Besides considering standard metrics related to performance (success rate and reward), we also consider safety properties that are particularly important in these domains (e.g., the agent does not collide with obstacles). In more detail, we employ formal verification ((Corsi et al., 2020)) to compute the percentage of input cases that cause violations of these properties. This is important to confirm our claim that Supe-RL based approaches correctly bias the exploration process in the direction of more robust policy regions with higher returns.

Results show that Supe-RL algorithms improve performance (i.e., training time, success rate, average reward), stability, and safety over value-based and policy-gradient DRL (Rainbow (Hessel et al., 2018), PPO (Schulman et al., 2017)) and ERL. Finally, we performed additional comparisons of Supe-RL with: (i) PDERL (Bodnar, 2020) to evidence the poor performance of previous mixed approaches when combined with value-based DRL; (ii) CEM-RL (Pourchot, 2019) in the aquatic scenario, to show the differences with a multi learner approach; (iii) ERL in standard continuous benchmarks (i.e., MuJoCo locomotion (Brockman et al., 2016; Todorov et al., 2012)), where results confirm the superior performance of Supe-RL.

## 2 BACKGROUND AND RELATED WORK

We formalize robotic navigation as a RL problem, defined over a Markov Decision Process, as described in recent DRL literature (Tai et al., 2017; Zhang et al., 2017; Wahid et al., 2019). DRL for robotic navigation focus exclusively on continuous action algorithms such as actor-critic DDPG (Lillicrap et al., 2015), TD3 (Fujimoto et al., 2018) and PPO (Schulman et al., 2017). Such methods

have been adopted following the idea that value-based DQN (Mnih et al., 2013) can not deal with high-dimensional action spaces. However, discrete value-based solutions typically result in shorter training time being more sample efficient, and show better performance even in continuous settings. In detail, Marchesini & Farinelli (2020b) shows that discrete DRL is a more efficient alternative to continuous DRL in robotic navigation. Moreover, Tavakoli et al. (2018) proposes an adaptation of Dueling DQN (Wang et al., 2016) with Double DQN (van Hasselt et al., 2016) that achieves competitive results in locomotion benchmarks (Brockman et al., 2016; Todorov et al., 2012). More recently, de Wiele et al. (2020) designed a DQN-based algorithm to handle enormous discrete and continuous action spaces. These researches further motivate our contribution in the design of a mixed approach that also works with value-based DRL.

**Evolutionary Algorithms** EA are an alternative black-box optimization characterized by three main operators (Fogel, 2006): generation, alteration, and selection. In detail, Montana & Davis (1989) evolves a population of $N$ individuals, each one represented by the network vector parameter $\theta$ (genome). Each $\theta_i$ ($i \in [0, .., N-1]$) is evaluated to produce a fitness $F(\theta_i)$, used by the selection operator to choose the best genome. For the EA component of Supe-RL, we consider a mutation-based GA that shown competitive performance over gradient-based DRL (Such et al., 2017).

**Combining EA and DRL** Following the trend of using EA as an alternative for DRL (Salimans et al., 2017), an emergent research field focuses on combining gradient-free and gradient-based solutions. In particular, ERL (Khadka & Tumer, 2018) considers an actor-critic DDPG agent (Lillicrap et al., 2015) and a concurrent EA training that generates a population of individuals, which are mutated and selected based on their fitness. The DRL agent is trained in parallel from the samples generated by both the training phases and it is periodically injected in the running population which is used to collect the training performance. The mutation function of ERL ensures that, in a certain number of episodes, the gradient-based policy outperforms its evolutionary siblings, introducing the gradient-based benefits into the population. Hence, biasing the selection process of the next generation and its performance. In their experiments, the authors highlight an efficient transfer of information between the two families of algorithms, outperforming DDPG in well-known locomotion benchmarks (Brockman et al., 2016; Todorov et al., 2012). However, both the introduction of all the experiences in the buffer and forcing the DRL agent to perform better than the EA population, bias the training and can cause detrimental behaviours. Inspired by ERL, several combinations have been proposed (Bodnar, 2020; Colas et al., 2018; Pourchot, 2019; Khadka et al., 2019). While GEP-PG (Bodnar, 2020) can be considered as a simplified version of a mixed approach, where a curiosity-driven approach is used to fill the buffer of the agent, Proximal Distilled ERL (PDERL) (Bodnar, 2020) addresses the EA component of ERL, introducing novel operators to compensate for the simplicity of the genetic representation (as investigated by Lehman et al. (2018), where authors address destructive behaviors of biologically-inspired variation operators applied to neural networks, which causes catastrophic forgetting). However, as detailed in Section 3.1, our genetic evaluation is used to soft update the DRL agent only in the case of performance improvement, hence it does not show such catastrophic forgetting. We also mention CERL (Khadka et al., 2019) and CEM-RL (Pourchot, 2019) as they are extensions of ERL for distributed training (which we do not consider here) with multiple active learners, which leads to non-negligible overhead (nonetheless, Section 4.2 reports an additional experiment in our continuous task with CEM-RL, to provide a more heterogeneous overview of the superior performance of Supe-RL). These works share a common baseline as they all rely on actor-critic DRL and are built on the insights of ERL, which is the most closely related to Supe-RL. Hence, we choose ERL for complete performance comparison. Section 4.2 also shows a comparison with PDERL, to further highlights the poor performance of previous approaches when combined with value-based DRL.

Finally, we use formal verification to support our claims on the beneficial effects of our genetic component into the policy. We report in Appendix A a brief description of the considered methodology.

## 3 SUPE-RL

The main insight of Supe-RL is to soft update a DRL agent towards better policy regions, enabling the combination with any DRL algorithm. In detail, we combine a mutation-based GA (Such et al., 2017) with two DRL algorithms[1]: (i) Rainbow (Hessel et al., 2018) as a value-based algorithm for

---

[1]We evaluated a variety of different learning algorithms for both action domains. Among Rainbow, PPO, DDPG, and TD3, we chose the best-performing ones.

the discrete task and (ii) PPO (Schulman et al., 2017) as policy-gradient one for the continuous scenario, giving rise to two algorithms named SGRainbow and GPPO.

A typical Supe-RL based algorithm (Appendix B provides a general pseudocode) proceeds as follows: the weights of a DRL agent $drl_a$ (referred as genome or $\theta_a$ interchangeably), are initialized with random values and, as in a standard training setup, $drl_a$ starts to collect experiences interacting with its environment. Such experiences are stored in a buffer $R$ to train the network. Periodically (every $GE$ episodes), Supe-RL makes a genetic evaluation by generating a population of children, each one characterized by a different genome. Weights $\theta_a$ are used to create the $N$ individuals applying noise to the parameter vector: $\theta_a + mut_p n$, where $n \sim \mathcal{N}(0, mut_v)$ and $mut_p$ is the mutation probability. In contrast, the mutation function of ERL multiplies the randomly chosen weights by $\mathcal{N}(0, mut_v)$. Such mutations act in a similar fashion of a dropout layer, biasing the DRL agent to perform better than the evolutionary population in the long term. In more detail, ERL authors mutate 10% of the network weights in each episode (or epoch), multiplying them by $\mathcal{N}(0, 0.1)$ (plus a mutation with standard deviation 10 or a reset mutation in a small percentage of cases). Given that their evolutionary component is running in parallel with the gradient-based agent, we noticed that the weights in the population tend to 0, hence causing a detrimental behavior. The population and a copy of $\theta_a$ are then independently tested over a set of evaluation episodes which shares the same goals, to find the overall best performing individual $\theta^{best}$ based on the fitness (the fitness definition is domain-specific; in the case of navigation, it is computed as the number of targets the agent reaches over the evaluation episodes). In this phase, we can also store a portion of diverse experiences in the buffer $R$ used by $drl_a$, to further exploit the population-based component. Finally, if the selected genome belongs to one of the children, $drl_a$ weights are updated towards the mutated version and the training phase continues with the new weights. In contrast, if the best score belongs to $drl_a$, the training phase, which was running in parallel, continues. Since the evaluation does not require any interaction among the population, we instantiate an independent copy of the environment for each $N + 1$ population component in a separate thread and test them in parallel, drastically reducing the overhead for the $drl_a$. The multi-thread nature of the Unity game engine makes this parallel testing phase straightforward and particularly efficient. This approach has both the advantage of search a better-performing policy exploiting GA mutations (similar to noisy exploration (Fortunato et al., 2017)) and enrich the replay buffer with new diversified experiences. As detailed in our empirical evaluation and the experiments with formal verification tools (Section 3.1), our genetic evaluation leads to safer policies and a significant reduction in training time, with Supe-RL resulting approximately two times faster than ERL in the same scenario.

It is important to mention that both Supe-RL and previous mixed approaches are especially designed for scenarios in which it is possible to parallelize the learning process. Hence, when the training phase is performed on real physical systems, it is not possible in general to evaluate the population in parallel. This could significantly increase the convergence time.

Crucially, in contrast to previous mixed approaches, Supe-RL based algorithms are designed to improve the performance of $drl_a$ as our combination schema does not bias the choice of the better-performing children in the long term. In the worst-case scenario, the main agent is always the best genome in the population and does not improve the current policy, hence a Supe-RL based training will match the performance of the chosen DRL algorithm.

## 3.1 Value-Based and Policy-Gradient Implementations

Robotic navigation allows evaluating Supe-RL in a variety of scenarios (e.g., discrete and continuous action spaces). Hence, our two tasks present different characteristics (e.g., static and dynamic environments, sparse and dense rewards). In this section, we present the value-based and policy-gradient implementation of Supe-RL, combined with a mutation-based GA (Such et al., 2017).

### 3.1.1 SGRainbow in Discrete Indoor Navigation

We consider a Turtlebot3[2] indoor environment with obstacles, using a discrete action space and a dense reward function.

---

[2]https://www.turtlebot.com/

**Environment description** Target goals randomly spawn in the scenario and are guaranteed to be obstacle-free. The reward $r_t$ is structured as two sparse value in case of reaching the target $R_{reach} = 1$ within error $\mu = 5cm$ from the target goal, or crashing $R_{fail} = -1$ which terminates an episode (resetting the robot to its starting position). A dense part is used during the travel: $\omega(d_{t-1} - d_t)$, where $d_{t-1}$, $d_t$ indicate the euclidean distance between the robot and the goal at two consecutive time steps and $\omega = 10$ is a multiplicative factor.

**Network architecture:** the input layer contains 19 sparse laser scans, sampled in $[-90, 90]$ degrees in a fixed angle distribution and the target position (expressed in polar coordinates); a similar setting is used in Tai et al. (2017); Long et al. (2018). We did explore other encodings for the problem (e.g., adding linear velocities as output) but a higher complexity of the problem causes longer training times with negligible improvements. We performed multiple trials on different network sizes and seeds (Chen & Chang, 1996) and the outcome led us to use two $relu$ hidden layers with 64 neurons each and 5 $linear$ nodes to encode the output angular velocities $[-90, -45, 0, 45, 90]$ $deg/s$.

**Methodology for SGRainbow:** the genetic evaluation of the value-based agent presents challenges due to the instability of the training algorithm (which is a known drawback of DQN-based algorithms) and the poor scalability on high-dimensional action spaces. Nonetheless, recent work shows the benefits of such solutions (Marchesini & Farinelli, 2020b; Tavakoli et al., 2018; de Wiele et al., 2020) in these challenging settings, further motivating the requirement for a mixed approach that works also with value-based DRL. Here we will discuss only the elements of the algorithm which are relevant to SGRainbow, referring the interested reader to (Hessel et al., 2018) for further details on Rainbow. We developed different approaches to update the $drl_a$ with the genome of the better performing child. We firstly switch $drl_a$ with the child, soft updating only the target network (originally developed for DDQN van Hasselt et al. (2016)) to approach the new weights. We tried different settings for the target network, but a soft update with $\tau' = 0.1$ showed us better performance. This method however leads to an unusual optimizer choice, which is crucial considering the high-variance of a value-based algorithm. In particular, the widely adopted Adam optimizer (Kingma & Ba, 2014) with its self-adaptable learning rate typically requires minimal hyper-parameter tuning but, after a switch of the $drl_a$ model with a better genome, a drop of performance occurs. We motivate this as the current learning rate of Adam is "balanced" for the old $drl_a$ weights and requires time to adjust its value for the new unexpected and better-performing model. For this reason, this first application, which we refer to as GRainbow, required to tune an SGD optimizer for this scenario, decaying its initial learning rate from 0.1 to 0.001 based on the current success rate of the model. As shown in Section 4, this approach already outperformed both the Rainbow algorithm and the GA. The tuning for SGD requires several trials and is one of the main limitations of GRainbow. To address this, we considered Tessler et al. (2019), where authors improved DDQN stability by copying the agent model to the target when the former performs better. Hence, we flipped our approach by soft updating $drl_a$ towards the best genome (using $\tau' = 0.1$) and switching the target network with such genome. This enabled us to use the Adam optimizer, improving the performance of GRainbow. We refer the interested reader to Appendix C of the Supplementary Material for a performance comparison between GRainbow with SGD and Adam.

Finally, we further improved both training performance and stability, exploiting the soft update (Lillicrap et al., 2015) technique for both networks. We refer to the resultant algorithm as Soft GRainbow (SGRainbow). In particular, we soft update both the $drl_a$ agent and target models towards the weights of the best performing child to smooth the transition of the networks towards the better-mutated policy. The update rule for the $drl_a$ networks is then: $\theta_a = \tau'\theta_a + (1 - \tau')\theta^{best}$. We tried different values for $\tau'$ (results in Appendix C), obtaining the best performance with $\tau' = 0.3$. We believe that these periodical slight changes in the $drl_a$ policy, that simulate a gradient step towards better network weights, are the core mechanism that allowed Supe-RL to work and improve the performance of value-based Supe-RL, while also reducing the variance across different runs. A useful side message is that using SGD seems better than Adam when $drl_a$ "jumps" due to target network instabilities, as Adam works better if the transition of the target network is more stable. Finally, results in Section 4 introduce part of the experiences of the genetic evaluation into the same prioritized buffer of the $drl_a$ (Appendix C contains an ablation experiment on the influence of these diversified experiences with SGRainbow).

### 3.1.2 GPPO in Continuous Aquatic Navigation

We consider an aquatic drone navigation task characterized by dynamic waves, with a continuous action space and a sparse reward function. Sources of the aquatic simulator as a novel DRL task and a video with an overview of the environment are available at `tinyurl.com/y22xh43c`.

**Environment description:** The aquatic drone is a differential drive platform, where a continuous action is mapped to the motor power. The drone receives a sparse reward $r_t$ structured as two values in case of reaching the target $R_{reach} = 1$ within error $\mu = 5$ cm from the target goal, or reaching timeout $R_{fail} = -1$ which terminates an episode (resetting the robot to its starting position).
**Network architecture:** the input layer contains the target position expressed in polar coordinates with respect to the drone and the pitch of the boat. It is possible to retrieve these values using the GPS and the compass of the boat. Our initial evaluation for the network size led us to use two $tanh$ hidden layers with 32 neurons each. Finally, two $tanh$ output nodes encode each motor velocity (multiplied by a constant value to obtain our velocity limit).

**Methodology for GPPO:** in contrast to the value-based implementation, the stability of on-policy PPO does not present issues related to the Adam optimizer. Hence, we considered the same soft update strategy adopted with SGRainbow. Trivially we did not use experiences of the best child due to the on-policy nature of the algorithm. For further details on the implementation of the PPO algorithm, we refer to (Schulman et al., 2017).

## 4 Empirical Evaluation

The goal of our empirical evaluation is to investigate whether Supe-RL approaches combine the benefits of GA with both value-based and policy-gradient DRL while maintaining minimal overhead for the training. Data are collected on an i7-9700k, using the implementation of Section 3.1. We considered the same set of hyper-parameters (reported in Appendix D) for the baselines and Supe-RL based approaches. Our ERL implementations refer to (Khadka & Tumer, 2018) for the evolutionary component, while we use Rainbow and PPO for the gradient-based part. In order to get reproducible and consistent results when comparing different algorithms, the random seed is fixed across a single run of every algorithm (because there may exist a sequence of targets that favor a run or a better network initialization), while it varies in different runs. As a consequence, a certain run of every algorithm executes the same sequence of targets and initializes the networks with the same weights.

All the trained models, except for the value-based ERL (ERL-R) and the GA, are able to navigate generalizing: starting and target position, and velocity. The Turtlebot3 lidar allows navigating in unknown environments with different obstacles, while the boat maintains similar performance in different wave conditions. For each graph, we report the mean and standard deviation of ten statistically independent runs, considering (i) success rate: how many successful trajectories are performed; (ii) total reward. Results are smoothed over one hundred episodes.

To collect significant data in the evaluation of navigation performance, we chose a set of targets reachable by every model. Table 1 resumes the results for the discrete (Rainbow, SGRainbow) and the continuous (PPO, ERL, GPPO) tasks, considering only models with acceptable performance (i.e., we did not include GA and ERL-R). For clarity, in the aquatic task we collected the dense reward used in the discrete one. Hence, rewards will be similar

Table 1: Performance in the evaluation phase

| Model | Reward | Step | Time (s) |
|---|---|---|---|
| Rainbow | 36.2 | 370 | 42 |
| SGRainbow | 39.1 | 269 | 29 |
| PPO | 4.5 | 98 | 27 |
| ERL | 4.6 | 96 | 27 |
| GPPO | 4.8 | 89 | 22 |

(agents navigate towards the target, collecting positive rewards), but time and number of steps (i.e., trajectory length) differ significantly ($\approx 31\%$ and $\approx 37.5\%$ for SGRainbow over Rainbow; $\approx 18\%$ and $\approx 8\%$ for GPPO over PPO and ERL, respectively). In both tasks, Supe-RL outperforms DRL algorithms and ERL in every considered metric.

### 4.1 Value-Based Evaluation

Here we compare the GA, Rainbow, ERL-R, GRainbow, and SGRainbow. As previously discussed, Figures 2A, B show that a direct combination of ERL with the value-based algorithm can not cope

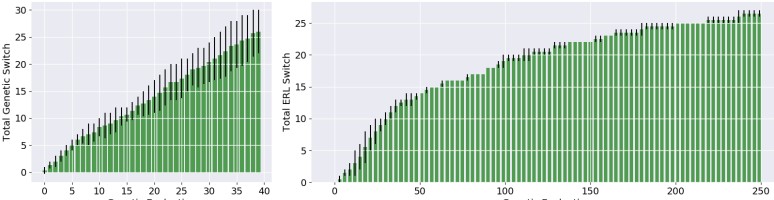

Figure 2: Left: indoor navigation (GA, Rainbow, ERL-R, GRainbow, SGRainbow). Right: aquatic navigation (GA, PPO, ERL, GPPO). (A, C) Average success rate. (B, D) Average total reward.

Figure 3: Left: cumulative number of Supe-RL genetic soft updates in continuous aquatic navigation. Right: cumulative number of injection of the DRL agent using ERL in the same environment.

the issues of such DRL approaches, resulting in very poor performance. Given these results, we decided to perform an additional experiment with the improved version of ERL, PDERL (Bodnar, 2020), which introduces novel genetic operators to improve ERL robustness. Nonetheless, we conjecture that the detrimental behaviour of previous mixed approaches is related to their DRL injection pattern, rather than the simplicity of the genetic approach (the goodness of a simple genetic representation is evident in our results and in Such et al. (2017)). In contrast, our periodical genetic evaluation with a soft update strategy simulates a gradient step towards a better policy while, in a worst-case scenario, we do not update the DRL agent, avoiding detrimental behaviours. Our results (detailed in Section 4.2) confirm the superior performance of PDERL over ERL in both our environments. However, PDERL still provides inferior performance compared to Supe-RL based approaches and shares the poor performance of ERL when combined with the value-based algorithm. In further detail, Figure 2A shows that even GRainbow with the SGD optimizer, where the best genome fully substitutes the main DRL agent, outperforms Rainbow (i.e., $80\%$ of success rate over $60\%$). Moreover, the soft genetic update further improves the performance while reducing the variance and SGRainbow reaches a performance of $90\%$ success rate in about 2000 epochs that correspond to 60 minutes of training (in contrast, Rainbow reached $60\%$ success rate in similar training time). Furthermore, the standalone GA was not able to cope with the complexity of the task, where the algorithm needs to generalize the navigation while exploiting the laser values to avoid obstacles.

## 4.2 POLICY-GRADIENT EVALUATION

Here we compare PPO, an adapted version of ERL with PPO, and GPPO. Figures 2C, D show that also in the continuous domain, Supe-RL based method offers better performances considering our evaluation metrics. In detail, GPPO reaches over $98\%$ of average success rate in about 1300 epochs that correspond to 110 minutes of training, while PPO, similarly to ERL, was able to reach $\approx 82\%$ of average success rate in $\approx 1700$ epochs (160 and 210 minutes of training, respectively). Furthermore, as reported in Table 1, GPPO uses a lower number of actions compared to PPO and ERL, which translates into shorter paths and travel time for the drone towards the same target.

In this task, we also compared the efficiency of our genetic evaluation in finding better policies with respect to ERL. Figure 3 shows the cumulative number of injections (ERL) and evaluations (Supe-RL) (x-axis), over the successful ones (y-axis) through the training, to reach similar performance. We compare injections and evaluations as they represent the approaches of ERL and Supe-RL to improve the policy, respectively. Results show that our mutation schema finds more often better-performing policies, in contrast to ERL where successful network injection occurs more rarely (ERL requires $400\%$ more injection trials, i.e., 250 over 25, to match Supe-RL performance, i.e., $98\%$ success).

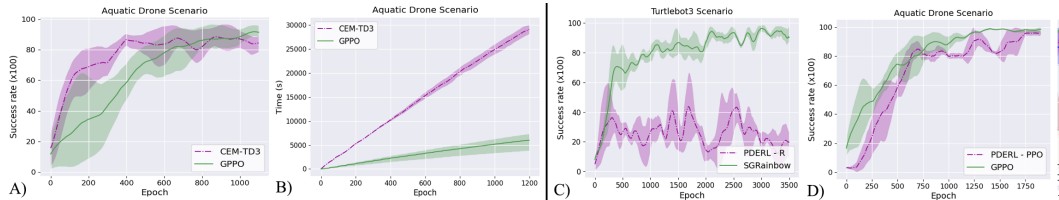

Figure 4: Performance of Supe-RL, PPO and ERL in MuJoCo benchmarks: (A) Reacher-v2; (B) HalfCheetah-v2; (C) Hopper-v2; (D) Ant-v2.

Figure 5: Comparison with CEM-RL in aquatic navigation: (A) average success rate; (B) average training time. Comparison with PDERL in the navigation tasks: (C) Value-Based implementation of PDERL in the discrete task; (D) Policy-Gradient implementation of PDERL in the continuous task.

**Evaluation in Standard Benchmarks** We performed additional experiments on Reacher-v2, HalfCheetah-v2, Hopper-v2, and Ant-v2 MuJoCo tasks (Brockman et al., 2016; Todorov et al., 2012) with GPPO and ERL. In particular, we considered the same specifics for data collection detailed in Khadka & Tumer (2018), hence we report the same performance metrics instead of a success rate. Our ERL implementation with PPO returned comparable results to the original ones presented in Khadka & Tumer (2018); Pourchot (2019); Khadka et al. (2019) (data were collected using the same hardware, GPPO parameters, and averaged over 5 runs). Crucially, Figure 4A, B, C, D shows that our Supe-RL based algorithm has comparable or better performance across the considered tasks. Furthermore, 4C highlights the detrimental behaviour of ERL.

**Comparison with CEM-RL and PDERL** To confirm that previous mixed approaches can not be directly combined with value-based approaches, due to their injection pattern rather than their genetic representation, we performed an additional experiment with PDERL (Bodnar, 2020). Such approach addresses the poor genetic representation of ERL, improving its performance and robustness. Figure 5C, D shows the collected data in the discrete scenario, where we combined PDERL with our value-based baseline, and in the continuous task, where we combined it with the PPO baseline. We considered PDERL GitHub for the genetic component, and the same hyper-parameters and random seeds used in our previous experiments. Results confirm the superior performance of PDERL over ERL in both environments (i.e., improved success rate and reduced variance). Furthermore, the two parallel training phases do not provide a robust evaluation of the gradient-free population in the same set of tasks, hence, the resultant best agent does not represent an overall best policy for the task. Follows that PDERL shares the detrimental performance of previous mixed approaches when combined with value-based algorithms. Finally, as detailed in Section 2, the recent field of mixed approaches has been extended with distributed solutions that use a portfolio of active DRL learners, such as CERL (Khadka et al., 2019) and CEM-RL (Pourchot, 2019). This intuitively results in significant overhead in the training process. We decided to compare GPPO with the TD3 version of CEM-RL, as the authors claim it outperforms previous mixed approaches (to further confirm the overhead, Sec. 5.2.2 of CEM-RL (Pourchot, 2019) state that tests are on limited timesteps, due to computational demands). In contrast, Supe-RL uses one DRL learner and the population is only used for policy evaluation. We used the GitHub provided by the authors to test CEM-TD3 in the continuous scenario (aquatic drone) and Figures 5A, B show that, as expected, CEM-RL required significantly more time (250 minutes over 110). In particular, CEM-TD3 reaches 98% success rate (the performance of Supe-RL at epoch 1200) in approximately 600 episodes. However, it required 125% more wall-clock time with respect to the time required by Supe-RL to reach similar performance (data were collected using the same hardware and averaged over 5 runs).

Table 2: Verification results of: (top) indoor task; (bottom) aquatic task

| | Violation (%) | | | Time (s) | | | Memory (MB) | | |
|---|---|---|---|---|---|---|---|---|---|
| **Model** | $\Theta_{I,0}$ | $\Theta_{I,1}$ | $\Theta_{I,2}$ | $\Theta_{I,0}$ | $\Theta_{I,1}$ | $\Theta_{I,2}$ | $\Theta_{I,0}$ | $\Theta_{I,1}$ | $\Theta_{I,2}$ |
| Rainbow | 2.21 | 9.11 | 0 | 79.7 | 75.5 | 92.6 | 3.74 | 3.96 | 6.92 |
| SGRainbow | 0 | 4.75 | 0 | 66.7 | 74.1 | 80.5 | 2.18 | 2.91 | 4.1 |

| | Violation (%) | | Time (s) | | Memory (MB) | |
|---|---|---|---|---|---|---|
| **Model** | $\Theta_{A,0}$ | $\Theta_{A,1}$ | $\Theta_{A,0}$ | $\Theta_{A,1}$ | $\Theta_{A,0}$ | $\Theta_{A,1}$ |
| PPO | 0.9 | 1.2 | 3.4 | 124 | 0.1 | 5.8 |
| ERL | 0.5 | 0.7 | 3.4 | 3.4 | 0.1 | 0.15 |
| GPPO | 0 | 0.1 | 3.1 | 3.2 | 0.1 | 0.1 |

### 4.3 ROBUSTNESS OF SUPE-RL USING FORMAL VERIFICATION

An important result in both tasks is the limited variance shown by Supe-RL approaches, across runs with different network initialization seeds. Appendix E shows a more detailed analysis of our results, where Supe-RL based approaches seem to not suffer from detrimental network initialization seeds.

To further confirm our claims on the beneficial transfer of information of mixed approaches, we employ a formal verification tool Corsi et al. (2020), to verify the behavior of our trained models with respect to a series of safety properties. The idea behind this evaluation is to confirm that models trained with mixed approaches lead to more robust policies. Given the inferior performance of ERL, we also expect that Supe-RL based models will present fewer configurations that cause undesirable behaviors (i.e., violates the safety property). According to the navigation scenarios, we selected the following safety properties for the $I$ indoor and $A$ aquatic navigation tasks:

$\Theta_{I,0}$: If TurtleBot3 has obstacles too close on the right and on the front, it must turn left.
$\Theta_{I,1}$: If TurtleBot3 has obstacles too close on the left and on the front, it must turn right.
$\Theta_{I,2}$: If TurtleBot3 has obstacles too close on the front, it must turn in any of the other directions.
$\Theta_{A,0}$: If the aquatic drone has a target on the right, it must turn right.
$\Theta_{A,1}$: If the aquatic drone has a target on the left, it must turn left.

Table 2 shows violation percentage, computation time, and memory returned by the verification tool, to test our safety properties. In detail, models trained with mixed approaches (i.e., SGRainbow, GPPO, and ERL) present fewer violations in every considered property. Furthermore, there is also a significant improvement over the computation time and memory required by the verifier. This confirms our claims on the policy improvement of mixed approaches as they evaluate with a significant difference the output values, which translates into fewer bounds re-computations for the verifier. Crucially, the superior safety performance of Supe-RL based approaches compared to ERL, further motivate the introduction of our framework.

## 5 DISCUSSION

We presented Supe-RL, a novel mixed framework that exploits the robustness of population-based GA to improve value-based and policy-gradient DRL agents. We evaluate Supe-RL in two mapless navigation scenarios: an indoor navigation for a TurtleBot3 platform, an aquatic navigation task, and in locomotion benchmarks. Our empirical evaluation shows that Supe-RL significantly outperforms DRL baselines (Rainbow and PPO), the GA, and the recent ERL and PDERL, which also show poor performance when combined with value-based DRL. Crucially, Supe-RL is the first framework that combines GA and DRL in the field of value-based discrete methods. Furthermore, we exploited a formal verification tool to confirm the beneficial effects of mixed approaches in policy improvement. This evaluation confirms the superior performance of Supe-RL based approaches, that returned safer models. This work paves the way for several interesting research directions which include exploiting complex evolutionary mechanisms and different crossover techniques to further improve our framework as well as the possibility of extending Supe-RL to concurrently optimize the total reward and the desired safety properties.

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

## A    INTERVAL ANALYSIS FOR FORMAL VERIFICATION

We consider formal verification to support our claims on the beneficial effects of our genetic component on the policy. We briefly describe the main verification concept and refer the interested reader to the references provided. Recently, Wang et al. (2018); Singh et al. (2019) propose an extension of interval analysis for the verification of neural networks. Given a set of input bounds (area) and a safety property to verify, these methods propagate the input, layer by layer, to compute the bounds for each output node. Afterwards, they exploit a standard formalization from interval algebra (Moore, 1963): $y_0 = [a, b] < y_1 = [c, d] \Rightarrow b < c$ to prove or deny properties. In decision-making scenarios, such as our navigation, we can verify a property by analyzing the relationships among the computed output bounds. To simplify the understanding of this process, we visualize

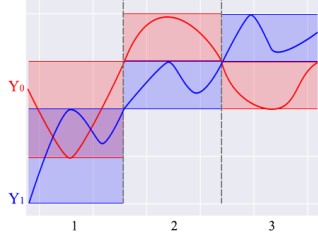

Figure 6: Bound analysis in formal verification tools.

it as a 2d graph in Figure 6 which shows the possible scenarios after computing the bounds for a decision-making task: (i) there is an overlap between output bounds; (ii) an output bound that satisfy the property is always higher than the others; (iii) an output bound corresponding to an action that violates the property is greater than the others. In the last two cases, we can conclude that the property is either verified or denied in the input set. However, the first case requires to re-compute the output bounds in smaller input subsets (Figure 6 shows the bounds on three smaller subsets on the x-axis) to obtain a more precise estimation that falls in one of the other verifiable cases.

In summary, by exploiting formal verification (and specifically Corsi et al. (2020)) we can measure the number of input configurations that would cause a violation of our safety properties. We measure this as the percentage of area that may cause a violation with respect to the entire input area (i.e., the violation percentage). The lower this number, the more robust is the policy (i.e., properties may be violated by a lower amount of input configurations). This translates, de facto, in an overall better policy as the trained network correctly evaluate with a significant difference the output values (output bounds have very different values), i.e., the action to choose. In detail, fewer bounds re-computation (i.e., fewer overlaps) means faster computation and less required memory.

## B    SUPE-RL PSEUDOCODE

We introduce a general pseudocode of Supe-RL based algorithms.

---
**Algorithm 1** Supe-RL
---
1: **Input:** size $N$ of the population $P$, periodicity $GE$ for the genetic evaluation, seed $s$, timeout $T$. Initialize network weights $\theta_a$, replay buffer $R$ and $env$ with seed $s$.
2: **for** episode = 1 **to** $\infty$ **do**
3:     **for** t = 1 **to** $T$ **do**
4:         Following the algorithm policy, select action $a_t$
5:         Execute action $a_t$, observe reward $r_t$ and state $s_{t+1}$, and Store $(s_t, a_t, r_t, s_{t+1})$ in $R$
6:     **end for**
7:     $env \leftarrow$ ResetEnvironment()
8:     Sample a random mini-batch of transitions from $R$
9:     Perform a gradient descent step, updating $\theta_a$
10:    **if** episode $\% \ GE == 0$ **then**
11:        **Start periodical genetic evaluation, in parallel**
12:        $P, Genvs \leftarrow$ GenerateChildren($N, \theta_a, env$)
13:        $Score \leftarrow$ GeneticEvaluation($P, Genvs$)
14:        $\theta^{best} \leftarrow$ Select best $\theta \in P$ based on $Score$
15:        $\theta_a \leftarrow$ GeneticUpdate($\theta_a, \theta^{best}$)
16:        **End periodical genetic evaluation**
17:    **end if**
18: **end for**
---

---

**Algorithm 2** Function GenerateChildren

---

1: **procedure** GenerateChildren($N$, $\theta_a$, $env$)
2:   $Children \leftarrow$ Initialize with $N + 1$ copies of $\theta_a$
3:   $Cenvs \leftarrow$ Initialize with $N + 1$ copies of $env$
4: **for** i = 1 **to** $N$ **do**
5:    **for** iteration = 1 **to** $mut_p * |Children[i]|$ **do**
6:      Randomly sample indices $r, c$ for $Children[i]$
7:      $Children[i][r, c] + = \mathcal{N}(0, mut_v)$
8:    **end for**
9: **end for**
10: Return $Children$, $Cenvs$

---

## C  ADDITIONAL SGRAINBOW EXPERIMENTS

In this section, we show the performance of GRainbow with SGD and Adam and the results of the tuning of $\tau'$ parameter for SGRainbow. We also present an ablation experiment on the influence of the population experiences in the buffer of the DRL agent.

**Adam and SGD GRainbow** Tessler et al. (2019) improved the original DDQN evaluating the performance of the agent model over its target model. In particular, they show that a more stable training phase can be obtained replacing the target model with the agent weights when the latter statistically perform better than the former. Inspired by this approach, when our genetic evaluation finds a better child, the main agent is soft updated towards the best genome, and the latter substitutes the target model. As Figure 7A shows, this precaution allows us to use GRainbow with Adam and outperform the manually tuned GRainbow with SGD.

**Tuning $\tau'$ for SGRainbow** We further improved GRainbow performance, exploiting the soft update (Lillicrap et al., 2015) developed to improve the naive approach in DDQN which periodically substitutes the target with the agent model. We performed multiple trials with different seeds to find the best value of $\tau'$ for the soft genetic update function that characterizes SGRainbow. In particular, Figure 7B clearly shows that SGRainbow outperforms GRainbow and reports the average success rate of our training phases with $\tau' \in [0.15, 0.3, 0.6]$.

**Population Experiences in SGRainbow** Finally, given our best performing algorithm (i.e., SGRainbow with $\tau' = 0.3$), we performed an ablation experiment to test the influence of the experiences collected in the genetic evaluation phase in the main agent buffer. Figure 7C shows performance improvement with the introduction of the best child experiences.

## D  HYPER-PARAMETERS

**Genetic Evaluation** We use the same genetic hyper-parameters for both SGRainbow and GPPO for the genetic evaluation. In particular, the size $N$ of the population $P$ is set to 10 and the number of trials conducted in the evaluation phase to compute the fitness score ranges from 10 to 20 across tasks. The mutation probability of a network weight is $mut_p = \{0.75, 0.4, 0.1\}$ based on the current success rate of the $drl_a$ and a constant mutation value $mut_v = 0.1$.

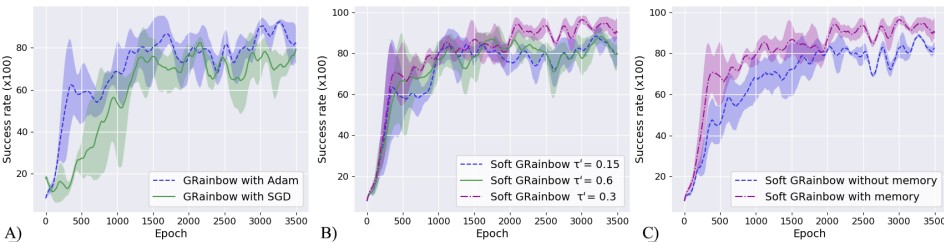

Figure 7: Additional value-based experiments: Average success rate for (A) SGD and Adam GRainbow; (B) Tuning of $\tau'$ for SGRainbow; (C) Ablation on population experiences in agent buffer.

**Rainbow, GRainbow, and SGRainbow** Here we discuss only the hyper-parameters of the algorithmic implementations which are relevant to GRainbow and SGRainbow, referring to the original papers for further details (notice that the three approaches share the same hyper-parameters). Based on our experiments, we decided to use a Priority Experience Replay (Schaul et al., 2016) buffer of size 30000 with a batch size of 64, $\alpha = 0.6$, $\beta_{start} = 0.4$ and $\beta_{increment} = 0.0005$. The soft update of the target network (Lillicrap et al., 2015) for Rainbow is performed with $\tau = 0.01$. Furthermore, given the well-structured reward function, we notice that a simple $\epsilon - greedy$ exploration strategy with $decay = 0.99$ and $\epsilon_{min} = 0.02$ led to a faster training phase compared to the introduction of noisy exploration in the network (Fortunato et al., 2017). Finally, we noticed that a genetic evaluation collects on average 30000 total transitions, and adding a random 10% of these transitions into the priority buffer (i.e., $trans_p = 0.1$), showed the best results.

**PPO, GPPO, and ERL** First, PPO presents two versions and we choose to implement the clipped one, which was recommended in the original work (Schulman et al., 2017), to which we remind the interested reader for further details. We choose to use a buffer of 256 elements with mini-batches of size 64 for the considered algorithms, while we used the original ERL (Khadka & Tumer, 2018) for specific algorithm parameters.

## E    ROBUSTNESS OF SUPE-RL

A more detailed analysis of our results reveals that the high variance of GA, PPO, and ERL is due to a specific seed that results in a pathological performance of the approach (a similar phenomenon occurs for Rainbow). In particular, Figure 8A, B, C, D report the success rate and total reward of the only bad initialization seed that causes GA, Rainbow, PPO, and ERL to perform poorly, compared to the previous runs. On the other hand, GPPO, GRainbow, and SGRainbow do not present this issue, reaching similar results to the other training phases. Even if this analysis is not as relevant as previous results, as it is not averaged over multiple seeds, we do believe that this is an important aspect of our framework and will need further investigation. Crucially, Figure 9A, B, C, D show that SGRainbow, and GPPO perform better than other approaches even in the selected runs without such detrimental seed.

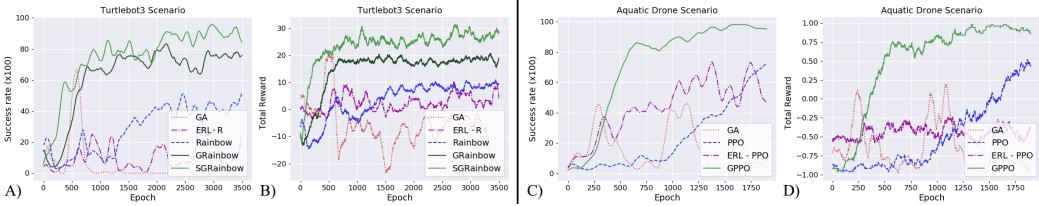

Figure 8: (A) Average success rate and (B) total reward considering only the simulation with the bad initial seed in the indoor navigation. (C) Average success rate and (D) total reward considering only the aquatic simulation with the bad initial seed.

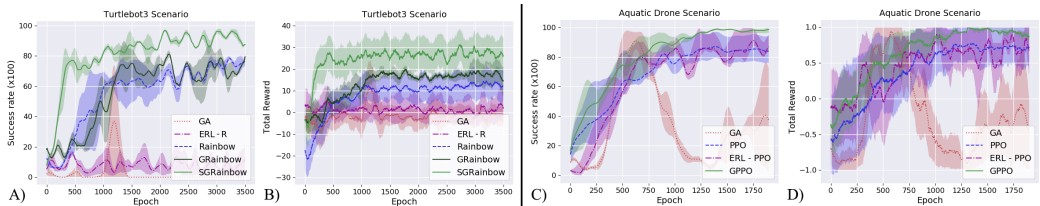

Figure 9: (A) Average success rate and (B) total reward when the bad initial seed is removed from the indoor simulations. (C) Average success rate and (D) total reward when the bad initial seed is removed from the aquatic simulations.

