# OpenReview forum: "Genetic Soft Updates for Policy Evolution in Deep Reinforcement Learning"
_ICLR.cc/2021/Conference — ICLR 2021 Poster_

### Official Review · AnonReviewer1 · 2020-10-29
**Evaluation is insufficient to reveal the advantage of the proposed approach over existing ones.**

**Rating:** 6
**Confidence:** 3

**Review:**

Aiming at exploiting the benefits of population based policy optimization and policy gradient, this paper proposes a novel framework that combines these two techniques. The authors claim that the previously proposed frameworks that combines evolutionary approaches and policy gradient approaches uses actor critic approaches in the policy gradient part, and that it is important to be able to incorporate value-based methods instead of actor critic approaches since the value based approach sometimes outperforms the actor critic approaches. The proposed framework is designed to be capable of combining evolutionary policy search approaches with ANY deep reinforcement learning algorithms. The framework is simple and relatively easy to combine with different deep reinforcement learning algorithms. However, in two instantiation of the proposed framework presented in this paper, the authors tune the details for each of these two. For one cases the authors obviously exploiting the structure of the combined reinforcement learning algorithm (existence of the target network and update the target network only), which is not possible in general.

The weakest point of this paper is the evaluation. The above mentioned two instantiations are evaluated only on a single task for each. It is definitely not sufficient to evaluate the generality of the framework. Moreover, the proposed approach is not compared to existing frameworks combining evolutionary approaches and policy gradients. The only existing approach is ERL, where its actor critic part is replaced so as to have the same RL approach as the proposed framework. Such a replacement is fine to evaluate the goodness of the framework itself. However, since ERL is not designed to be combined with other approaches than policy gradient, it is easy to imagine (and also mentioned in the introduction) that the performance will drop severely. To show the advantage of the proposed framework, the authors should compare the performance with the original ERL, and also more recent variants such as CEM-RL and CERL.

---

> ### Author Response · Authors · 2020-11-11
> **Re: Evaluation is insufficient to reveal the advantage of the proposed approach over existing ones.**
>
> We thanks the reviewer for the comments. However, we believe there are two important misunderstanding with our evaluation, which we clarify in detail below. Please refer to the revision.
>
> - Regarding the statement "For one case the authors obviously exploiting ..., which is not possible in general". It is correct that we present the proposed framework in two instantiations: one with value-based approaches (Rainbow) and one with policy-gradient ones (PPO). As detailed in Section 3.1.1, however, the presented value-based approaches do not update only the target network as stated by the reviewer.We further clarify this point: we first present GRainbow in two declinations: one that uses SGD and switch the drl agent with the better performing child (and then also updates the target network). The second version, that uses Adam, flips this approach by soft updating the drl agent towards the better performing child (and then updating the target network with a full copy of the child). Finally, our best performing approaches (SGRainbow) performs a soft update of both the agent networks. This combination strategy that soft updates the drl agent is also maintained in the policy-gradient PPO version that does not present any target network (Section 3.1.2). Hence, Supe-RL does not always require target networks. Nonetheless, we believe that our implementation with a value-based algorithm (that uses target networks) is important as many state-of-the-art approaches consider target networks (e.g., DDPG (Lillicrap et al., 2015), TD3 (Fujimoto et al., 2018), Rainbow (Hessel et al., 2018), SAC (Haarnoja et al., 2018)).
>
> - The misunderstanding that we believe mostly influenced the negative evaluation of our work, is related to the statement "The weakest point of this paper is the evaluation...". In the following points, we clarify this point:
>     1) First, Supe-RL is compared with both the gradient-based (i.e., Rainbow and PPO) and gradient-free (i.e., GA) baselines that we considered. We also compare our approach with a previous hybrid approach (i.e., ERL) in the main paper.
>     2) One of the main contributions of our work is that Supe-RL enables the combination with value-based approaches, addressing the limitation of previous hybrid frameworks. As noticed by the reviewer and in the main paper, ERL is not designed to be combined with other approaches than policy gradient. Hence, it is correct that in Section 4 we show a value-based version of ERL (to show the limitation of such approach); but we also evaluate Supe-RL with a policy-gradient algorithm in our continuous navigation task. Here ERL is combined with a policy-gradient algorithm and, in fact, it does not show performance drop.
>     3) Furthermore, in the first submission, Appendix C presented a comparison with PDERL in both our navigation tasks, to confirm that previous hybrid approaches are indeed not designed to work with value-based algorithms.
>     4) Regarding the comparison with CEM-RL and CERL, we discussed in Section 2 that these approaches uses a portfolio of active gradient-based learners, hence they are mostly related to the family of distributed approaches (that we do not consider in this work). Nonetheless, it is not correct that we do not compare Supe-RL with such approaches as in Section 1 we remind the reader to Appendix A (which is now included in Section 4.2 of the main paper), that shows a comparison between our GPPO and CEM-RL (we considered CEM-RL as it showed better performance with respect to CERL (Pourchot, 2019)).
>     5) Finally, as noticed by the second reviewer, it is not correct that we evaluate Supe-RL in a single task for each. As detailed in Sections 1 and 4, Appendix B (which is now included in Section 4.2 of the main paper) presented a comparison in four additional MuJoCo tasks.
>
> Considering the comments on the evaluation and given the additional page for the rebuttal phase, we revised the paper (especially Section 4) moving the previously mentioned appendices that contained the comparisons with PDERL, CEM-RL and an evaluation in MuJoCo tasks, in the main paper.

---

> > ### Author Response · Authors · 2020-11-23
> > **The requested evaluation was moved in the main paper. Additional comments on the validity of our ERL implementation.**
> >
> > Following the insightful discussion with AnonReviewer2 in which we were able to clarify most of the reviewers' concerns and lead to a significant change of score, we would like to achieve a similar constructive discussion also with AnonReviewer1, which did not provide us any feedback since our rebuttal.
> >
> > We would like to remind the reviewer that the additional experiments (i.e. in standard locomotion benchmarks) and comparisons with other previous approaches (i.e., CEM-RL, PDERL) were already present in the appendices of our first submission. In the current revision, we moved such experiments in Section 4.2 of the main paper to make them more visible to the reader.
> >
> > Furthermore, the other main point of AnonReviewer1 is related to our implementation of ERL. To this end, we provided an additional detailed answer to AnonReviewer2 that we summarize here for more clarity.
> > The reviewer's claim that "ERL is not designed to be combined with other approaches than policy gradient" is indeed correct. However, ERL-PPO is based on the PPO algorithm, which is a policy gradient approach. Hence, the PPO implementation does not limit ERL.
> > Our results in Section 4.2 (Figure 4) clearly show the validity of our ERL-PPO implementation as the experiments in the standard MuJoCo benchmarks return comparable performance with the ERL implementation proposed by the original authors (Khadka & Tumer, 2018).
> >
> > We would like to remark that the discussion phase is about to close and we did not receive any feedback or update from AnonReviewer1, despite we clarified the reviewer's misunderstanding with the additional experiments/comparisons, and the ERL implementation.

---

> > > ### Comment · AnonReviewer1 · 2020-11-24
> > > **Thank you for the detailed comments.**
> > >
> > > Dear authors, Thank you for our detailed comments and sorry for my late reply. I overlooked the appendix of the original submission. There results in "Evaluation in Standard Benchmarks" (now in the main text) is rather convincing. Still a weakness is that Figure 5 ("Comparison with CEM-RL and PDERL") compares the proposed framework only on a single task for each algorithmic instance.
> > >
> > > I understand that the authors point is to make value-based approach to be successfully combined with evolutionary approaches. I already understood this point in my first review. However, I think this motivation is not very clearly mentioned in the main text. In the introduction, I found "This is important as recent work (Matheron et al., 2019) shows the limitation of actor-critic in deterministic tasks, that in contrast can be effectively addressed with value-based methods" Are the tasks used in this paper examples of such cases? This point is still unclear to me.

---

> > > > ### Author Response · Authors · 2020-11-24
> > > > **Re: Thank you for the detailed comments.**
> > > >
> > > > Thank you for your feedback. First, since we did not receive any additional questions on our ERL implementation, which was one of your initial issues, we hope that our answer and the data match with the original implementation convinced the reviewer of the validity of the approach.
> > > >
> > > > Regarding your additional question on (Matheron et al., 2019), the tasks used in the paper are not examples of such a scenario, as Matheron et al. evidenced the issues of DDPG in the case of deterministic environments with sparse reward (which we do not consider). The reason why value-based algorithms are important is discussed in more detail in Section 2. In particular, Tavakoli et al. (2018), de Wiele et al. (2020) shows that value-based approaches can address high-dimensional action spaces with superior performance with respect to policy-gradient solutions. Furthermore, this is also clearly shown in Marchesini & Farinelli (2020), where authors consider a navigation task, similar to the one we consider in our work.
> > > >
> > > > The misunderstood with our results in the standard benchmark has been clarified, hence the fact that we outperform the considered baselines in such continuous (policy-gradient) tasks is yet a contribution. Such contribution is further exacerbated by our value-based implementation and the formal verification. We agree that we evaluated the value-based algorithms in a single navigation task, but we would like to remark the following important points:
> > > >
> > > > 1. The methodology behind the genetic component of Supe-RL is the same as the policy-gradient version and we outperform the considered baselines on six different tasks in total.
> > > >
> > > > 2. The navigation problem does not imply any assumption that could favor either the proposed framework or the baselines, providing a general task for our evaluation in both value-based and policy-gradient domains.
> > > >
> > > > 3. The analysis with formal verification in Section 4.3 confirms the benefits of our framework, as it shows the beneficial transfer of information, with Supe-RL resulting in overall safer policies.
> > > >
> > > > 4. The goal of Figure 5 is twofold and we integrated it in the work for the following reasons:
> > > >         - CEM-RL is a distributed approach and, as detailed in our paper, we reported the experiment for better clarity. Distributed approaches, however, are not related to single-agent ones such as Supe-RL, ERL, PDERL.
> > > >         - The single experiment with PDERL was also an addition to the paper, to highlight the important limitations of previous approaches when combined with value-based algorithms.
> > > >
> > > > Hence, we believe that our contributions, which have also been recognized by the other reviewers, are still valid. After this constructive discussion that brings the additional experiments to the attention of the reviewer, and clarified his/her doubts on the ERL implementation, we hope for a revised evaluation of our work.

---

### Official Review · AnonReviewer3 · 2020-10-29

**Rating:** 6
**Confidence:** 4

**Review:**

This paper introduces another combination of an Evolutionary Strategy (ES) with RL, which basically consists of running an RL agent which every GE episodes alternates to run an evolutionary algorithm iteration based on the current policy as the parent of the population. If none of the individuals generated are better than the parent, the RL continues as it was before the genetic operations, otherwise the RL agent parameters are soft updated towards the value of the best individual.
The paper introduces a simple but effective approach which tries to obtain the benefits of both the ES and RL methods, making sure that the genetic operations do not  have a negative impact in the RL process. The contribution is interesting and fair, although the authors give considerable attention to the navigation problems used for validation,  which environments are also presented in this paper in the sections of the method and the results. Whereas in the title, abstract and first two sections the paper is completely focused only in the proposed hybrid learning method.
- Why not focusing on the more conventional/standard tasks (approached in the appendices) for evaluating the learning methods in the main paper, and leaving the proposed navigation tasks for the appendices, for a reader it would be more intuitive when analysing results of deeply well known studied problems. With this, the paper could focus more on the method, because for instance, Section 3 is dedicated considerably to stress the navigation tasks, while the method is indeed more general and not specialized in this kind of tasks. On the other hand, the content of Appendix E would be more valuable in the main paper.
- In Section 3, it is mentioned "Such mutations act in a similar fashion of a dropout layer, biasing the DRL agent to perform better than the evolutionary population in the long term". It is not clear why it is better, what is the foundation of this statement? Authors could add references or present a deeper discussion about it. If the reason is given based on the experiments of this work, this statement fits better the experiments section, or conclusions.
- It is important to stress more that the method is especially intended for scenarios in which it is possible to parallelize the learning process. In cases of learning with physical systems, it is not possible to compute the genetic operators and the population evaluation in parallel, this could significantly increase the convergence time, which could be unfeasible.

- For SGRainbow in Section 3.1.1 it is mentioned "...as GRainbow, required to tune an SGD optimizer for this scenario, decaying its initial learning rate from 0.1 to 0.001 based on the current success rate of the model". How is this decay computed? it seems it is a function of the success rate, however for tasks that do not have this metric, how could this be done?
- Why the statistics are taken with only 5 runs in Section 4? this does not seem to be enough, additionally the tasks do not look extremely complex to think that running more experiments is unfeasible (this is also why analysing main results based on the standard tasks is more straightforward).
- Regarding the results of Table 1, it is not clear what are exactly time and number of steps, since normally they represent the same information. It is said steps have to do with trajectory length,  but what does length mean? it could be either trajectory duration, or distance of the trajectory.
- In general the results look positive and show an improvement over the baselines and previous methods combining the two worlds. It would be interesting to mark in the learning curves, the times in which the genetic operation was executed, in order to observe its effect through the convergence.

---

> ### Author Response · Authors · 2020-11-11
> **Re: Official Review of Paper3101 by AnonReviewer3**
>
> We thank the reviewer for his/her positive feedback. We will clarify the reported doubts sequentially. Please refer to the revised version of the paper.
>
> - Regarding the navigation problem, we did introduce such robotic tasks from the abstract ("Experiments in robotic applications and continuous control benchmarks") onwards. In Section 1, we motivate the navigation ("Our evaluation focuses on mapless navigation...") as it represents a well-known problem in the DRL robotics community (Zhang et al., 2017; Wahid et al., 2019; Marchesini & Farinelli, 2020). Furthermore, in Section 2 there is a paragraph on DRL navigation literature and the importance of value-based approaches.
>
> The main reason for this choice, however, is related to its versatility:
>   1) we were able to design both a discrete and a continuous action space environment for our evaluation, hence using an application of real interest.
>   2) the novel usage of formal verification to support the beneficial claims of Supe-RL requires the design of safety properties (as detailed in Section 4.3). Such design is straightforward in the case of discrete and continuous navigation (i.e., check the output values relationship in the value-based approach, and the motor velocity in the continuous one), while it is not trivial in standard tasks. As an example, in the case of the HalfCheetah continuous task, it is complex to design a property that controls the safety of the six actuators output. This is also the reason why formal approaches do not consider standard DRL benchmarks for their evaluation, but more suitable tasks, i.e., the ACAS Xu models (Wang et al., 2018a, Wang et al., 2018b). Nonetheless, to provide a more heterogeneous evaluation of our approach, we evaluate Supe-RL in the standard continuous benchmarks considered by previous hybrid approaches (as noted by the reviewer).
>
> - Regarding the statement that in ERL (Khadka & Tumer, 2018) "Such mutations act in a similar fashion of a dropout layer...". The foundation of this statement relates to the mutation function considered in their work. Considering authors' implementation and hyperparameters, they mutate 10% of the network weights in each episode/epoch by multiplying them by a normal distribution with mean 0 and standard deviation 0.1 (plus a mutation with standard deviation 10 or a reset mutation in a small percentage of cases). Given that their evolutionary component is running in parallel with the gradient-based component, we noticed that the weights in the population tend to 0 causing a detrimental behavior.
> - We agree with the reviewer that we should stress the fact that Supe-RL is not intended for online learning on physical systems, as in such scenario we can not parallelize the process. However, we do believe this is not a severe limitation of our contribution as the same consideration holds for both distributed approaches (that uses a multitude of active learners) and previous hybrid methods (that also considers a population of agents).
> - GRainbow details in section 3.1.1 state that it requires to tune a decaying learning rate for the SGD optimizer. The decay is indeed in function of the success rate, and in tasks that do not have this metric, it could be designed in function of the cumulative reward. However, as detailed in the same section, the main contribution in combining value-based DRL and our GA is related to SGRainbow, which uses Adam and does not require such limitation (e.g., the complex and time-consuming tuning).
> - Regarding the statistics collected in 5 runs, we chose the same number of runs considered in previous hybrid approaches (Khadka & Tumer, 2018, Bodnar, 2020) both for the evaluation in our navigation tasks (Section 4) and the evaluation in standard locomotion benchmarks (Section 4.2). Furthermore, other state-of-the-art approaches such as DDPG (Lillicrap et al., 2015), are also evaluated over 5 runs, while PPO (Schulman et al., 2017) is evaluated over only 3 runs.
> - Regarding Table 1, the time expressed in seconds is the wall-clock time to reach the set of targets, while the number of steps is the number of performed actions. Hence, this is related to the agent path (which can be expressed in meters). Notice that, as detailed in such evaluation, we chose a set of targets reachable by every model, hence time and steps are related. However, in the case of collisions, the recovery procedure of the agent will influence only the time.
> - In the current version of the paper, a reader has to interpret the effects of the genetic operator by combining the information in Figures 2 and 3. As noticed by the reviewer, this visualization does not directly show the effects of the genetic operation. However, the learning curves are expressed over multiple runs and a genetic operator does not always influence the gradient-based agent. Hence, it would be possible to express this in separate graphs for each run, marking the only epochs where the genetic operator influences the learning.

---

> > ### Comment · AnonReviewer3 · 2020-11-23
> >
> > Thanks for the detailed reply, I think the authors have taken the time to revise the topics related to the concerns carefully.
> >
> > - Yes you have mentioned those problems are important when giving the context of the problem and applications, however, the original structure had a loose focus since it was introducing the learning method and the problem of the application in navigation, which is not the only target problem of the learning method. Using the space for introducing that problem shortened the space for giving details of the proposed algorithm. Additionally, since the environment itself is not known, it is not easy to assess the complexity of the task before actually working with it, which is not the case in standard environments.
> >
> > - Thanks for the explanation, as mentioned before, this explanation should be included or cited in the paper, otherwise, this statement feels too strong and no support.
> >
> > - It is not necessarily bad to have some limitations, but it is always good to highlight them. Although you have argued a previous point saying that "Experiments in robotic applications and continuous control benchmarks"  is mentioned in the paper, but since robots are physical systems, this is not a minor detail, right? otherwise, you are pointing out the motivation of robotics in one discussion but avoiding it in the other. If this limitation holds for all these kinds of methods, maybe this is not the direction to follow in these kinds of applications. Motivating a method based on robotic problems requires to some extend validation on real platforms or analysis regarding its scalability
> >
> > - Clear.
> >
> > - As researchers, we study the design of experiments, and we know that processes which are not completely deterministic require statistics to be understood. The fact that famous researchers do different practices for particular reasons does not mean that designing experiments or the concepts of statistics are not valid anymore. Something that is not perfect does not give reasons to repeat things that can actually be improved. If some reviewers overlooked something, it does not mean it is correct, and that this  should be done that way from now on.
> >
> > -  Clear.
> >
> > - Clear

---

> > > ### Author Response · Authors · 2020-11-23
> > > **Re: Official Comment by Paper3101 AnonReviewer3**
> > >
> > > Thank you for the feedback. Following your suggestions, we uploaded a revised version of the manuscript that includes:
> > > - The explanation of the claim "Such mutations act in a similar fashion of a dropout layer..." in Section 3
> > > - A brief paragraph in Section 3 that clarifies the limitation of Supe-RL and previous approaches in learning with physical systems (i.e., it is not possible to parallelize the learning process)
> > > - Minor formatting revisions to include previous changes without exceeding the limit of 9 pages
> > >
> > > Following your additional comment on the design of experiments, we agree that collecting data on additional runs will further improve the collected statistics.
> > > However, the discussion period is about to close and we will consider this for the final version of the paper (in case of acceptance) or in a future revision of our works.

---

> > > > ### Comment · AnonReviewer3 · 2020-11-24
> > > >
> > > > I agree it is fine to include the additional runs in the final version in case of acceptance, hopefully, results are not so different, but regardless of how they are, they would be useful for more reliable conclusions. With only few runs, it would not be a surprise that reproducing an experiment will get different results with respect to what is presented in this paper.

---

### Official Review · AnonReviewer2 · 2020-11-02
**Interesting idea - reservations about the technical rigor of the manuscript**

**Rating:** 7
**Confidence:** 4

**Review:**

The paper introduces Supe-RL that intermingles off-policy reinforcement learning with periodic beam search operations. The method makes a greedy selection between the rl-solution that it had and the best produced by the beam search using Polyak update to update the incumbent rl-solution if the one suggested by beam search is better. Experiments in robotics simulation benchmarks show that Supe-RL improves over prior hybrid methods.

An extensive review of related literature is included that is extremely helpful in situating the work amongst the broader field. Visuals like Figure 1 also really hep the user in understanding the core concepts presented.

The paper could be clearer with more copy-editing and technical rigor. Some technical terms are used loosely and ancillary claims made without justification which serve to confuse the reader. For example, the author claims (Page 1) that gradient-free approaches have poor generalization skills. I am not quite sure if this is correct as gradient-free EA/GA has been an industry standard as a general-purpose black-box optimizer.

The author also claims that ERL uses an ES. I have read the ERL paper in the past and from memory know this to be incorrect. Verifying with the ERL paper, it seems to use an EA, not an ES. This is crucial as an ES is a finite-difference approximation of the gradient at heart and would be rather redundant with a gradient-based learner. EA on the other hand belongs to an entirely different family of algorithms and resembles natural evolution more closely.

The author also uses the word redundancy in a confusing way. First, the author claims that redundancy in the population leads to diverse experience (page 4). I am not sure how this would work. Robustness is certainly a contributing factor to redundancy but diversity? I think the author is trying to convey that a population allows for a spread-out search across multiple promising directions/points in the search space concurrently thus helping diversity. I am not sure if redundancy is the correct word to describe this feature.
Similarly, the author mentions enriching the (replay) buffer with new redundant experiences (Page 4 Section 3)? That seems like an oxymoron.
Page 5 reads “Finally, results in Section 4 considers the redundancy offered by the population to introduce part of the experiences of the genetic evaluation into the same prioritized buffer of the drla” – I am not quite sure how redundant experiences would add value?

The author refers to Polyak updates as soft updates and makes it a central component of the method’s novelty. However, these kinds of updates with a target network is a fairly standard tool in most modern policy gradient algorithms i.e., Soft-Actor Critic (SAC). The use of the word “soft update” is also a bit confusing as I thought this indicated the use of a modified Bellman backup with a Boltzmann policy representation (similar to Soft Q-learning) at first. It is unfortunate that the namespace is overloaded.

If my understanding is correct, the mutated solution (neural network policy) from a round of EA directly replaces the DRL solution? This is generally a risky thing to do. Gradient-based methods often rely on the weights to lie within specific distributions (ranges of magnitudes) for effective gradient computation and update. This is particularly true if the network uses non-linear activation functions that can easily saturate (dying ReLU since the authors use ReLU). However, EAs have no such requirement and the weights can often blow up to big numbers particularly for a non-linear function approximation. Is there a safeguard to avoid this scenario?
I believe the nuances behind sensitivities of the optimizer (SGD and ADAM) that the author mentions (plus the fact that Polyak updates help) are symptoms of this fundamental issue (transferring policy weights with a nonlinear function approximator directly from a GA to a gradient-based learner).

Overall, the authors present an interesting idea. However, the novelty is marginal compared to earlier methods and far too many technical liberties are taken with the manuscript that make it difficult to absorb as a reader.

---

> ### Author Response · Authors · 2020-11-11
> **Re: Interesting idea - reservations about the technical rigor of the manuscript**
>
> Thanks for acknowledging our effort in positioning our work in related literature and for your valuable comments about technical rigor. We have revised the paper to correct the usage of misleading terms (such as "redundancy" and "ES"). Please refer to the revision. Below, we sequentially address the concerns in detail and clarify the novelty of Supe-RL.
>
> - Regarding the claim on page 1 "gradient-free approaches have poor generalization skills", we rephrased this last part of the sentence as we agree with the reviewer that it was misleading. However, the fact that EAs suffer from high sample complexity and often struggle to solve high dimensional problems that require optimization of a large number of
> parameters is correct (and it is also claimed by previous hybrid approaches such as ERL (Khadka & Tumer, 2018)).
>
> - We agree with the reviewer that our usage of the term "Evolutionary Strategies (ES)" is not correct and we revised it in the paper. As pointed out by the reviewer, ESs are a finite-difference approximation of the gradient (Salismans et al., 2017) and, according to prior work (Khadka & Tumer, 2018; Pourchot et al. 2019) they are a class of the broader family of Evolutionary Algorithms (EAs), which also includes the Genetic Algorithm that we consider in our work.
>
> - We agree with the reviewer that page 4 has an incorrect usage of the word "redundancy" in explaining that the redundancy in the population leads to diverse experience and in mentioning that we enrich the buffer with new redundant experiences. Hence, we revised the usage of the word "redundancy" as in some parts of the paper it was confusing.
>
> - We refer to Polyak updates as soft updates as DDPG (Lillicrap et al., 2015), to the best of our knowledge, was the first DRL algorithm to use Polyak updates and, in such work, authors refer to this methodology with the term "soft update".
>
> Besides the revision of these terminologies, we believe that there is some important misunderstanding with our methodology that contributes to the negative evaluation of our work.
>
> The reviewer's comment "the mutated solution (neural network policy) from a round of EA directly replaces the DRL solution..." is not correct. As detailed in our two implementations (Sections 3.1.1 and 3.1.2) our best performing implementations of Supe-RL do not replace the DRL solution. Both SGrainbow and GPPO perform a Polyak update towards the weights of the best-mutated solution with a factor τ' = 0.3. Only the first implementation of GRainbow (the one that uses SGD) naively replaces the DRL agent with the mutated one. However, we recognized in the paper that this causes several issues (as an example, the tuning of SGD) and we presented two better-performing alternatives that do not perform such detrimental replacement.
> Furthermore, as pointed out by the second reviewer, we utilize several attention to make sure that our genetic operations do not have a negative impact on the RL process. In more detail:
>     1) the Polyak update towards the mutated solution simulate a gradient step towards better network weights. The size of such step is bounded by the τ' parameter that we deeply analyzed in Appendix C.
>     2) As detailed in Section 3, we designed the evaluation episodes for the population to cover a wide variety of the behaviors that are required for the desired policy. In our experiments, this allows to obtain a good estimation of the overall performance of the population and the policies.
>     3) We update the drl agent only when one of the mutated children performs better than him. Furthermore, as detailed in Appendix D, we encourage the exploration in the proximity of the current drl policy, by scaling the mutation values.
>
> As noticed by the other reviewers, we want to highlight that our framework allows the combination with any DRL solution. As evidenced by the poor performance in Section 4, previous hybrid methods do no allow a combination with value-based algorithms. The importance of value-based approaches is motivated in detail in Section 2. Furthermore, we also introduce a novel usage of formal verification to support our claims on the beneficial effects of our genetic component. The collected data in Table 2 further motivate our contribution as results show that policies trained with Supe-RL present fewer property violations (i.e., safer policies). This also paves the way for several interesting combinations of Supe-RL with formal verification to design novel Safe DRL approaches that are especially important in robotic applications.

---

> > ### Comment · AnonReviewer2 · 2020-11-22
> > **Re: [Reposted]**
> >
> > I have read the other reviews and the author's rebuttal. The authors have really taken the feedback from all the reviewers to heart and integrated most of the suggestions into the manuscript. Accordingly, I am raising my score.
> >
> > In light of the reviews/rebuttals I have a couple for additional questions/feedback:
> >
> > 1. A Polyak update with tau=0.3 is close to replacing the solution. Normally, in SAC/DDPG tau is in the order of 1e-3. Comparatively, 0.3 is an exceptionally large value (orders of magnitude larger) whereby ~5 of these chained updates would be a ~99% replacement of the incumbent policy. While this does not seem to be a fatal problem in the settings tested here, I would urge the authors to consider this more closely and add a detailed discussion for broader applicability.
> >
> > 2. In Fig 2c and 2d, one of the baselines is ERL-PPO. As far as I am aware, ERL is designed to work with an off-policy policy gradient learner. However, PPO is on-policy. This prompts me to wonder if this is a fair mix. If the PPO is sampling data from the ERL's central buffer, this would violate the on-policy assumptions since that data is cumulative across generation + being generated by a population of behavior policies with different parameters - thus making it extremely off-policy. Would an off-policy pg algo like SAC/TD3 be a better fit alongside ERL?

---

> > > ### Author Response · Authors · 2020-11-22
> > > **Re: "Re: [Reposted]"**
> > >
> > > Thank you for the feedback. Now that we clarified the miscommunication, we will further clarify both the validity of ERL-PPO and your additional question in the following points:
> > >
> > >     - Reviewer 1 comment "ERL is not designed to be combined with other approaches than policy gradient" is indeed correct, and this further motivates the contribution of our approach (as it also enables the combination with value-based methods). However, ERL-PPO is based on the PPO algorithm, which is a policy gradient approach. Hence, the PPO implementation does not limit ERL.
> > >     Our results in Section 4.2 (Figure 4) clearly show the validity of our ERL-PPO implementation as the experiments in the standard MuJoCo benchmarks return comparable performance with the ERL implementation proposed by the authors.
> > >
> > >     As in our GPPO implementation (detailed in Section 3.1.2), for ERL-PPO we did not use the experiences of the best child with ERL-PPO due to the on-policy nature of the algorithm. Regarding your other question, it is possible that an ERL-SAC implementation would return better results with respect to the original ERL, which is based on DDPG (but this would also probably results in better performance for the Supe-RL version). However, as detailed in page 4 (footnote 1), among PPO, DDPG, and TD3, we chose the former as it showed better performance in our evaluation domains.
> > >
> > >     2. We agree that the typical value for tau in the literature is in the order of 1e-3. However, these tau values are used in DRL algorithms to update the target network that has to slowly track the learned policy (Lillicrap et al., 2015), which is different from our usage.
> > >
> > >     In the proposed approach, we exploit the Polyak update to "move" the current DRL agent toward one of its slightly mutated versions that showed better performance in the same set of evaluation episodes (i.e., the best performing children after our genetic evaluation). To this end, we performed an initial experiment to choose the value for this parameter (Appendix D), obtaining the best results with tau' = 0.3.
> > >     The usage of such a large value is possible as the entire population is generated from the DRL agent weights using slight mutations, and such population (and a copy of the agent) is evaluated on the same set of episodes. Hence, we are indeed performing a large update of the agent, but towards a better version of itself.
> > >
> > >     Furthermore, as detailed in Section 3, the genetic evaluations (that lead to the update with tau'=0.3) are performed periodically. This allows the updated DRL agent to continue its learning process between two chained updates.

---

> > > > ### Comment · AnonReviewer2 · 2020-11-23
> > > > **Re:**
> > > >
> > > > The authors have sufficiently responded to most of my concerns. Accordingly, I am raising my score.

---

### Author Response · Authors · 2020-11-17
**Updated version of the paper**

Thanks for the feedback from the reviewers. As the first reviewer finds some "loosely used" technical terms that could confuse the reader, while the third reviewer did not consider the experiments reported in the paper appendices, we made the following revisions in the updated manuscript (we refer the reviewers to our answers for individual important clarifications):

    - We revised the usage of the terms "Evolutionary Strategies" and "redundancy" in different sections of the paper (e.g., pages 1 and 4) and some misleading claims such as "gradient-free approaches have poor generalization skills".
    - We moved our experiments in the standard benchmarks (i.e., continuous locomotion), and the other comparisons with CEM-RL and PDERL from the appendices to Section 4.2 of the main paper.

Considering the aforementioned changes to the paper, we would like to discuss with the reviewers the current version of our work.

---

### Decision · Program_Chairs · 2021-01-07
**Final Decision**

**Decision:**

Accept (Poster)

**Comment:**

This paper proposes a deep reinforcement learning algorithm Supe-RL that combines model free RL with genetic updates. The idea is to periodically mutate and evaluate the actor and greedily choose the best performing child, and incorporate it in the main actor via Polyak averaging on a target policy network. The algorithm can be in principle combined with any gradient based deep RL method. Supe-RL was demonstrated by combining it with Rainbow and PPO and evaluated in navigation tasks as well as standard MuJoCo benchmarks.

Overall, the reviewers found the idea interesting and to have value to the RL community. The reviewers raised some questions regarding technical rigor, evaluations, and the choice of base DL algorithms. As is, I find this a slightly above borderline submission, and thus recommend acceptance. However, I would encourage the authors to test their method also with a state-of-the-art off-policy algorithms, such as TD3 or SAC, in continuous domains, to better calibrate its overall performance.